# Transcriptomics and proteomics reveal a cooperation between interferon and T-helper 17 cells in neuromyelitis optica

Agnieshka M. Agasing[1,2], Qi Wu[3], Bhuwan Khatri[4], Nadja Borisow[5], Klemens Ruprecht [6],
Alexander Ulrich Brandt[5,7], Saurabh Gawde[1,2], Gaurav Kumar[1], James L. Quinn[1,2], Rose M. Ko[1],
Yang Mao-Draayer [3], Christopher J. Lessard[4], Friedemann Paul[5,6] & Robert C. Axtell [1,2✉]

Type I interferon (IFN-I) and T helper 17 (TH17) drive pathology in neuromyelitis optica spectrum disorder (NMOSD) and in TH17-induced experimental autoimmune encephalomyelitis (TH17-EAE). This is paradoxical because the prevalent theory is that IFN-I inhibits TH17 function. Here we report that a cascade involving IFN-I, IL-6 and B cells promotes TH17-mediated neuro-autoimmunity. In NMOSD, elevated IFN-I signatures, IL-6 and IL-17 are associated with severe disability. Furthermore, IL-6 and IL-17 levels are lower in patients on anti-CD20 therapy. In mice, IFN-I elevates IL-6 and exacerbates TH17-EAE. Strikingly, IL-6 blockade attenuates disease only in mice treated with IFN-I. By contrast, B-cell-deficiency attenuates TH17-EAE in the presence or absence of IFN-I treatment. Finally, IFN-I stimulates B cells to produce IL-6 to drive pathogenic TH17 differentiation in vitro. Our data thus provide an explanation for the paradox surrounding IFN-I and TH17 in neuro-autoimmunity, and may have utility in predicting therapeutic response in NMOSD.

[1] Arthritis and Clinical Immunology Research Program, Oklahoma Medical Research Foundation, 825 NE 13th St, Oklahoma City, OK 73104, USA.
[2] Department of Microbiology and Immunology, Oklahoma University Health Science Center, 940 Stanton L. Young Blvd., BMSB 1053, Oklahoma City, OK 73104, USA. [3] Department of Neurology, University of Michigan Medical School, 109 Zina Pitcher Place, Biomedical Research Building Room 4258, Ann Arbor, MI 48109, USA. [4] Genes and Human Disease Research Program, Oklahoma Medical Research Foundation, 825 NE 13th St, Oklahoma City, OK 73104, USA. [5] NeuroCure Clinical Research Center and Experimental and Clinical Research Center, Max Delbrueck Center for Molecular Medicine and Charité Universitätsmedizin, Lindenberger Weg 80, 13125 Berlin, Germany. [6] Department of Neurology with Experimental Neurology, Charité Universitätsmedizin, Charitéplatz 1, Berlin 10117, Germany. [7] Department of Neurology, University of California, Irvine Hall, R105, 252 Health Sciences Rd: 4290, 92697 Irvine, California, USA. ✉email: Bob-Axtell@omrf.org

Type I interferons (IFN-I), which include IFN-β and the various IFN-α molecules, are a family of pleotropic cytokines known to have antiviral, antitumor, and immune-modulatory functions[1,2]. In autoimmunity and inflammation, IFN-I possess both pro- and anti-inflammatory functions, depending on the context of the pathology. IFN-β is a widely prescribed treatment for multiple sclerosis (MS)[3], yet, it consistently worsens disease in patients with neuromyelitis optica spectrum disorder (NMOSD)[4,5]. The molecular mechanism behind the dual function of IFN-I in neuro-inflammatory diseases is currently unknown.

NMOSD is an autoimmune inflammatory disorder of the central nervous system primarily affecting the optic nerves and the spinal cord. In all, 60–90% of NMOSD patients have circulating IgG antibodies against the astrocyte water channel protein, aquaporin 4 (AQP4)[6]. Recent studies have also reported that a subgroup of NMOSD patients have autoantibodies against myelin oligodendrocyte glycoprotein (MOG)[7]. Several studies have shown that NMOSD patients have elevated TH17 signatures[8–10] and have increased frequency of relapses when treated with IFN-β[4,5,8,11,12].

The animal models of autoimmune CNS inflammation, collectively called experimental autoimmune encephalomyelitis (EAE), can be initiated by the adoptive transfer of myelin-specific TH17 or TH1 cells (TH17-EAE or TH1-EAE)[9,13–15]. We and others have found that TH17-EAE and TH1-EAE have strikingly different pathologies that reflect NMOSD and MS, respectively. Like NMOSD, EAE induced with TH17 cells manifests with severe optic neuritis, involves neutrophil infiltration into the CNS and has elevated levels of IL-17[9,13]. Even more striking are the differential effects of IFN-β treatment on TH17-EAE and TH1-EAE[14]. TH17-EAE mice had increased paralysis and increased inflammatory cell infiltration in the spinal cords when treated with IFN-β. Conversely, IFN-β treatment of TH1-EAE mice significantly reduced paralysis and inflammation in the CNS. These observations position TH17-EAE models as useful tools to study how IFN-I and TH17 drive pathology in diseases such as NMOSD.

The cooperative effects of TH17 cells and IFN-I in NMOSD and TH17-EAE were unexpected observations. The prevailing theory is that IFN-I inhibits the differentiation of TH17 cells and it has been speculated that the efficacy of this therapy in MS is achieved by inhibiting the function of the TH17 pathway[16–18]. This paradox represents a major knowledge gap in the field of neurology.

In this study, we perform biomarker studies in NMOSD patients and experiments in mice with TH17-EAE to resolve this paradox. Our data suggest the mechanism by which IFN-I contributes to the pathogenicity of TH17 cells is through the induction of IL-6 in B cells.

## Results

### RNA profiles stratify NMOSD based on IFN-I signatures. We performed whole-blood RNASeq to determine the transcriptional signatures that are associated with NMOSD disease compared with healthy controls. In our cohort of NMOSD patients, 62% of patients were on rituximab, 21% were on non-B-cell-depleting therapy and 17% of patients were not on disease-modifying therapy at blood draw (Supplementary Table 1). Rituximab-treated patients would skew our analysis of differentially expressed genes (DEGs) towards genes affected by B-cell depletion and away from genes associated with NMOSD etiology. In fact, when compared with healthy controls, we found that Rituximab-treated NMOSD (NMO-Ritux) patients had many more downregulated genes compared with NMOSD patients who

were not on Rituximab (NMO-Other Tx) or who were untreated (NMO-Untreated) (Fig. 1a–c, Supplementary Data 1–3). In order to determine gene expression signatures that are associated with NMOSD regardless of the therapy, we determined which DEGs were shared between NMO-Ritux, NMO-Other Tx, and NMO-Untreated. We found that 27 DEGs were shared between NMO-Ritux, NMO-Other Tx, and NMO-Untreated (Fig. 1d). Furthermore, using the Ingenuity Pathway Analysis software and the INTERFEROME database[19], we identified that 25 of the 27 shared DE genes were IFN-I-inducible genes (Fig. 1d). We found no statistical difference between the transcriptomes of NMOSD patients seropositive for AQP4 autoantibodies (AQP4-IgG$^+$) or MOG autoantibodies (MOG-IgG$^+$) and both had elevated expression of the IFN-I gene signatures compared with healthy controls (Supplementary Fig. 1).

In lupus, there is an association between the expression of IFN-I signature genes and variations in clinical features[20]. Therefore, we sought to determine whether IFN-I signatures can distinguish clinical differences in the NMOSD population. We found that hierarchal clustering of the 25 IFN-I genes (identified above) grouped NMOSD patients into two distinct subsets, patients with high IFN-I signatures (IFN-high) and patients with low IFN-signatures (IFN-low) (Fig. 1e). Patients on Rituximab, patients on other treatments, and untreated patients were represented in both, IFN-high and IFN-low groups (Fig. 1e).

### Proteomic signatures in IFN-high and IFN-low NMOSD. We next determined which inflammation-related protein biomarkers were associated with the IFN-I transcriptional signatures. We used a multiplex approach (OLINK) to assess the levels of 91 inflammatory proteins in the IFN-high patients and IFN-low patients compared with healthy volunteers. Using multivariate analysis of variance, we found that 26 inflammatory proteins were significantly elevated (with adjusted $p$ values of <0.05 and Log2FC > 0.5) in the IFN-high NMOSD patients compared with healthy controls (Fig. 1f, Supplementary Data 4). In comparison, only three proteins were elevated in the IFN-low NMOSD patients compared with healthy controls (Fig. 1f, Supplementary Data 4). As expected, we found that chemokines induced by IFN-I (CXCL9, CXCL10, CXCL11, MCP-3/CCL7) were elevated in the IFN-high patients but not in the IFN-low patients. We also found that IL-17A, the prototypic TH17 cytokine, and CCL20, a chemokine that promotes TH17 trafficking into inflamed tissue, were elevated in the IFN-high patients but not the IFN-low patients. Finally, we observed that IL-6 was among the most elevated proteins in the IFN-high patients (Fig. 1f). These data show that patients with high IFN-I also display elevated levels of serum IL-6 and proteins associated with the TH17 pathway.

### Blood markers are associated with disability in NMOSD. Next, we examined whether IFN-I transcriptional signatures were associated with clinical features in NMOSD patients. Strikingly, we found that IFN-high NMOSD patients had significantly higher scores in the expanded disability status scale (EDSS) as compared with IFN-low NMOSD patients (Fig. 1g). However, the two groups did not differ in terms of relapse rates, age, and autoantibody status to AQP4 or MOG antigens (Fig. 1h–j). We also assessed the utility of serum proteins to stratify patients based on EDSS. Here, we found that MCP-3 and IL-6 were significantly elevated in patients that had high EDSS scores compared with patients with low EDSS scores (Fig. 1k, l).

In addition, we assessed whether specific effector T helper subsets in PBMCs correlated with disability. For this, we obtained a collection of PBMC samples from a cohort of untreated

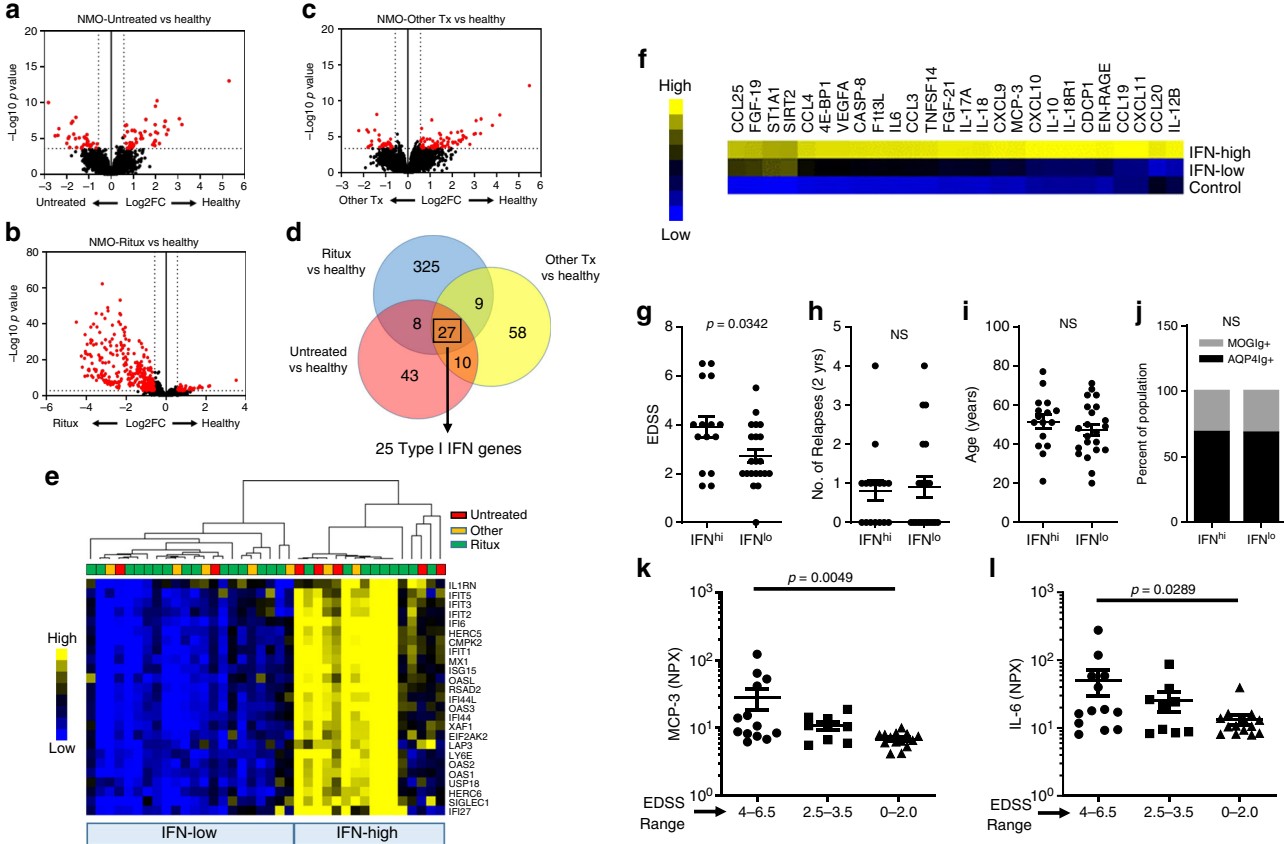

**Fig. 1 NMOSD patients stratify into two groups based on IFN-I gene expression.** RNA profiles of **a** untreated patients (NMO-untreated; $n = 7$), **b** Rituximab-treated patients (NMO-Ritux; $n = 24$) and **c** patients on other therapies (NMO-Other Tx; $n = 7$) were compared with healthy volunteers ($n = 18$). **d** Venn diagram of differentially expressed genes of the NMO-Untreated vs healthy, NMO-Ritux vs healthy, and NMO-Other Tx vs healthy. **e** Heatmap depicts relative levels of IFN-I genes in NMOSD patients (Red = NMO-Untreated, Yellow = NMO-Other Tx, Green = NMO-Ritux). Patients were stratified into two groups, IFN-low and IFN-high, based on IFN-I gene expression. Yellow represents relative high expression and blue represents relative low expression. **f** Heatmap depicts the differentially abundant serum proteins in IFN-high NMOSD ($N = 16$), IFN-low NMOSD ($n = 22$), and healthy controls ($n = 18$). Yellow represents relative high serum levels; blue represents relative low serum levels. Comparison of **g** disability (EDSS), **h** number of relapses 2 years prior to sample collection, **i** age, and **j** autoantibody status of IFN-high and IFN-low NMOSD patients. Two-tailed Student's $t$ tests and Chi-square tests were used to determine statistical significance. **k** MCP-3 and **l** IL-6 levels in NMOSD patients of different EDSS range (EDSS 4–6.5: $n = 15$, EDSS 2.5–3.5: $n = 9$, EDSS 0–2: $n = 16$). $P$ values were determined using two-tailed Kruskal–Wallis tests with multiple comparisons corrected by the Dunn's method. Bar graphs represent the mean and error bars are the S.E.M. Source data are provided as a Source Data file.

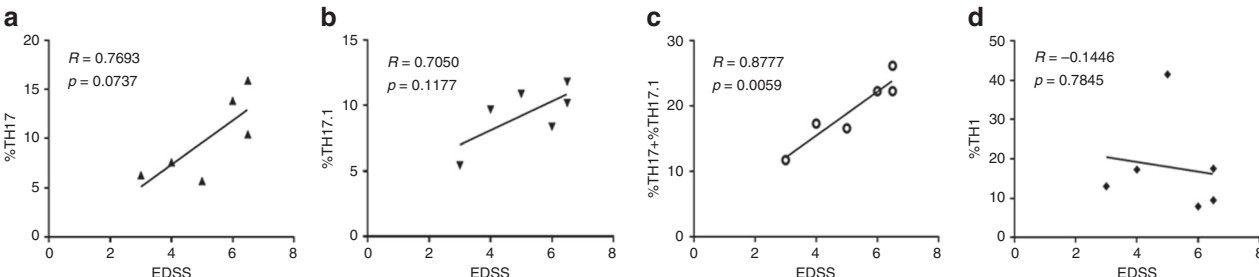

**Fig. 2 Correlation between TH17 and disability in NMOSD patients.** Correlations between EDSS and **a** %TH17, **b** %TH17.1, **c** %TH17 + %TH17.1, and **d** % TH1 cells in NMOSD patients ($n = 6$). Two-tailed Pearson correlations were used to determine statistical significance. $P$ values < 0.05 were considered significant and $P$ values > 0.05 were not significant.

NMOSD patients (Supplementary Table 2) and determined whether TH17 cells (CXCR3$^-$CCR6$^+$CD161$^+$), TH17.1 cells (CXCR3$^+$CCR6$^+$CD161$^+$), or TH1 cells (CXCR3$^+$CCR6$^-$CD161$^-$) correlated with EDSS. We found no clear correlation between TH17 and TH17.1 with EDSS (Fig. 2a, b). However, combined frequencies of TH17 and TH17.1 showed a significant positive correlation with EDSS (Fig. 2c). We did not observe a

positive correlation with EDSS and TH1 cells (Fig. 2d). In addition, we found that the percentage of TH17 cells, but not TH1 or TH17.1, was higher in the NMOSD patients compared to healthy volunteers (Supplementary Fig. 1).

Taken together, these data provide evidence that transcriptomic, proteomic, and Cell-Based-Assays can stratify NMOSD patients based on disability. Furthermore, these data suggest that

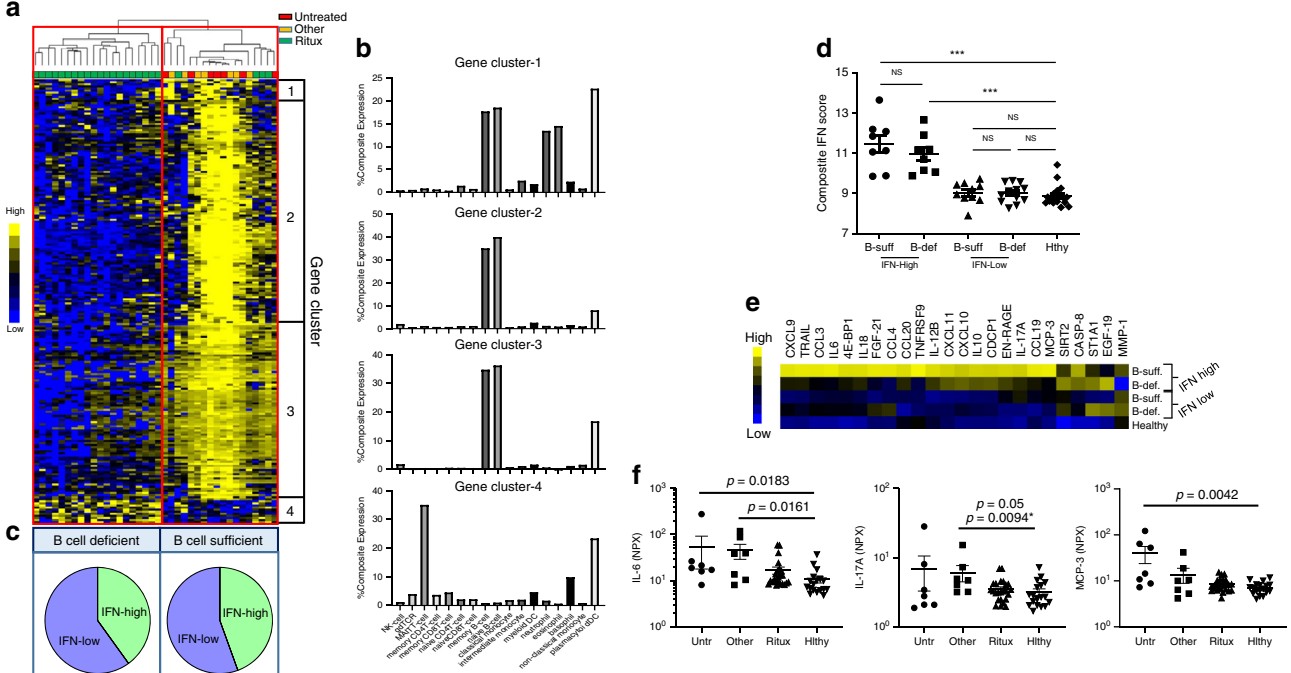

**Fig. 3 Effects of B-cell-depleting therapy (BCDT) in IFN-high and IFN-low NMOSD patients. a** Heatmap depicts the hierarchical clustering of NMOSD patients (Red = NMO-Untreated, Yellow = NMO-Other Tx, Green = NMO-Ritux) based on differentially expressed genes (DEGs) of patients treated with rituximab ($n = 24$) compared with patients not treated with rituximab ($n = 14$). Hierarchical clustering of the DEGs grouped genes into four clusters (Gene clusters 1–4). Yellow represents elevated gene expression and blue represents reduced gene expression. **b** Percent expression of genes in clusters 1–4 per cell type. **c** Stratification of NMOSD patients into two groups, defined as B-cell-deficient and B-cell-sufficient, based on gene expression and pie chart depicting the distribution of IFN-high and IFN-low NMOSD patients in the B-cell-deficient and B-cell-sufficient groups. **d** Composite IFN scores (average read count of IFN-I genes) of B-cell-sufficient IFN-high patients ($n = 8$), B-cell-deficient IFN-high patients ($n = 8$), B-cell-sufficient IFN-low patients ($n = 10$), B-cell-deficient IFN-low patients ($n = 12$) and healthy individuals ($n = 18$). **e** Heatmap indicating relative protein levels in IFN-high and IFN-low NMOSD patients from B-cell-sufficient or B-cell-deficient groups. **f** Serum protein levels of IL-6, IL-17A, and MCP-3 in untreated ($n = 7$), Other-treated ($n = 7$), and rituximab-treated NMOSD ($n = 24$) patients compared with healthy controls ($n = 18$). $P$ values were determined using two-tailed Kruskal–Wallis tests with multiple comparisons corrected by the Dunn's method or by a two-tailed Mann–Whitney test indicated with an *.

the cooperative effects of IFN-I, TH17, and IL-6 drive excessive CNS tissue damage that result in severe disability in NMOSD.

**Effects of B-cell-depleting therapy in NMOSD.** A popular therapy for NMOSD is B-cell depletion with anti-CD20 antibody[21]. In our cohort, 62% of the patients were on rituximab at the time of serum sample collection (Supplementary Table 1). To determine the transcriptional effects of rituximab treatment, we compared DEGs in patients treated with rituximab with patients not treated with rituximab (Supplementary Data 5). Since we found no significant difference in the transcriptomes of NMO-Untreated and NMO-Other Tx patients, we combined these patients for this comparative analysis. Based on the expression of DEGs, patients were clustered into two groups, which we defined as B-cell-deficient and B-cell-sufficient (Fig. 3a). We found that all patients in the B-cell-deficient group were on rituximab therapy. We also found that 14 of the 18 patients in the B-cell-sufficient group were not on rituximab or other B-cell-depleting therapies (Fig. 3a). Based on similar expression patterns in the population, the DEGs were divided into four gene clusters (Fig. 3a, b). Genes in clusters 1–3, which were reduced in the B-cell-deficient group, were determined to be predominantly expressed in B cells using the cell-specific RNA database (http://www.proteinatlas.org) (Fig. 3b)[22]. Conversely, genes elevated in the B-cell-deficient patients in cluster 4 were expressed in cell types other than B cells (Fig. 3b).

To determine whether B-cell depletion affects the IFN-I signature of NMOSD patients, we assessed the distribution of IFN-high and IFN-low NMOSD patients in the B-cell-deficient and B-cell-sufficient groups (Fig. 3c). The percentage of IFN-high and IFN-low NMOSD patients were similar in both B-cell-deficient and B-cell-sufficient patients (Fig. 3c). In addition, composite IFN scores, defined as an average read count of IFN-I genes, were not different between B-cell-deficient and B-cell-sufficient patients that were IFN-high or IFN-low (Fig. 3d). These data indicate that B-cell depletion with rituximab treatment does not impact IFN-I gene expression in NMOSD.

However, we did observe that serum protein profiles were significantly different in B-cell-deficient patients compared with B-cell-sufficient patients (Fig. 3e, Supplementary Data 6). Interestingly, we found that serum levels of IL-6, IL-17, and MCP-3 are highest in the IFN-high B-cell-sufficient NMOSD patients and were reduced in the B-cell-deficient IFN-high group (Fig. 3e, Supplementary Data 6). We also compared levels of IL-6, IL-17 and MCP-3 in NMO-Untreated, NMO-Other Tx, NMO- Ritux, and healthy controls. Serum IL-6 levels were elevated in NMO-untreated and NMO-Other-Tx patients, but not in NMO-Ritux patients; serum IL-17 was elevated in the NMO-Other Tx patients; and MCP-3 was elevated in the NMO-Untreated patients (Fig. 3f). These data suggest that B cells are a key cell type in elevating IL-6, IL-17, and MCP-3 in NMOSD patients.

Recent studies have suggested that the efficacy of B-cell depletion differs in AQP4-IgG+ and MOG-IgG+ NMOSD patients[23]. In our cohort, we compared annualized relapse rates in AQP4-IgG+ and MOG-IgG+ B cell-sufficient and B-cell-deficient patients. In the B-cell-sufficient patients, we found no

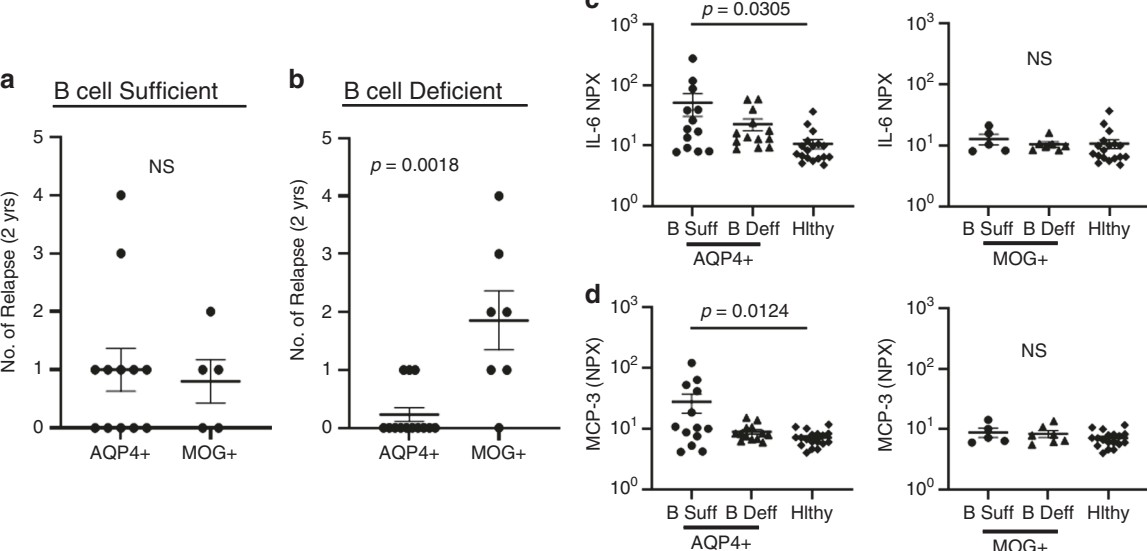

**Fig. 4 Effects of B-cell depletion on AQP4-Ig + and MOG-Ig + NMOSD. a** Comparison of relapse rates in AQP4-Ig + ($n = 12$) and MOG-Ig+ NMOSD ($n = 5$) who are B-cell-sufficient. **b** Comparison of relapse rates in AQP4-IgG+ ($n = 13$) and MOG-IgG+ ($n = 7$) NMOSD who are B-cell-deficient. $P$ values were determined using two-tailed Mann–Whitney tests. Serum **c** IL-6 and **d** MCP-3 levels in B cell-sufficient AQP4-IgG+ patients ($n = 13$), B-cell-deficient AQP4-IgG+ patients ($n = 13$), B-cell-sufficient MOG-IgG+ patients ($n = 5$) and B-cell-deficient MOG-IgG+ patients ($n = 7$) were compared with healthy controls ($n = 18$). $P$ values were determined using one-way ANOVA tests with multiple comparisons corrected by the Tukey's method. Error bars indicate the S.E.M. Source data are provided as a Source Data file.

differences in relapse rates between AQP4-IgG+ and MOG-IgG+ patients (Fig. 4a). In the B-cell-deficient group, we found that relapse rates were significantly higher in the MOG-IgG+ patients compared with AQP4-IgG+ patients (Fig. 4b). We next assessed serum IL-6 and MCP-3 levels in these patient groups and, strikingly, we found that both IL-6 and MCP-3 were elevated in the patients that were B-cell-sufficient and AQP4-IgG+ (Fig. 4c, d).

**IFN-I drives expression of IL-6 in human memory B cells.** The data above suggest that B cells are the major producers of IL-6 in NMOSD. We next questioned whether IFN-I drives IL-6 expression in B cells. In B-cell-sufficient NMOSD, we found significant positive correlations between IL-6 and IFN scores and between IL-6 and CXCL11 protein (Fig. 5a). In contrast, no correlation between IL-6 and IFN scores or between IL-6 and CXCL11 was seen in B-cell-deficient NMOSD patients (Fig. 5a). In healthy controls, no correlation was observed between IL-6 and IFN scores but there was a significant correlation between IL-6 and CXCL11 protein (Fig. 5a). Together, these correlations suggest that IL-6 is induced by IFN-I in B cells. To directly test this hypothesis, we isolated CD27− naive and CD27+ memory B cells from PBMCs of healthy donors. Both B-cell subsets were activated through CD40 and B-cell receptor in the presence or absence of IFN-β. We found that IFN-β stimulation did not alter CD80 and CD86 expression in naive B cells but significantly increased their expression in memory B cells (Fig. 5b, c). We also found that IFN-β stimulation did not alter IL-6 expression in naive B cells but significantly increased IL-6 expression in memory B cells (Fig. 5b, c). Thus, these data from patient sera and B-cell cultures from healthy donors provide strong evidence that IFN-I drives memory B cells to produce high levels of IL-6 in NMOSD.

**IFN-I exacerbates disease and elevates IL-6 in TH17-EAE.** Animal models are necessary to identify disease mechanisms which cannot be experimentally addressed in humans. We and others have reported that TH17-EAE mimics several features of NMOSD[9,13,14]. Like NMOSD, we previously identified that IFN-β treatment is not an effective therapy for TH17-EAE and instead exacerbates disease[14]. However, the mechanism by which this occurred was not identified. Consistent with our previous observations, IFN-β exacerbated paralysis (Fig. 6a) and increased the infiltration of immune cells and demyelination in the spinal cords of mice with TH17-EAE (Fig. 6b). Since IFN-I signatures and serum IL-6 were associated with increased disease burden in NMOSD patients (Fig. 1g, l), we tested if IFN-β treatment elevated IL-6 in mice with TH17-EAE. We found that IFN-β treatment significantly elevated serum levels of IL-6 (Fig. 6c). In addition, we found that numbers of T helper cells were elevated in the spinal cords of IFN-β treated mice (Fig. 6d). T-helper cells co-expressing IL-17 and granulocyte-macrophage colony-stimulating factor (GM-CSF) were also elevated in the CNS of TH17-EAE mice treated with IFN-β (Fig. 6e). We also measured B cells in the spinal cords of TH17-EAE mice treated with vehicle or IFN-β. We found that IFN-β treatment did not alter the number of B cells in the CNS of TH17-EAE mice (Fig. 6f). Our EAE data indicate that treatment of TH17-induced disease with IFN-I is associated with high levels of IL-6 and increased numbers of CNS infiltrating, inflammatory TH17 cells.

**Blocking IL-6 ameliorates IFN-I-treated TH17-EAE.** IL-6 is a potent inflammatory cytokine that is critical for the induction and pathogenic function of TH17 cells[24,25]. However, it has been reported that blocking IL-6 has little effect on TH17-EAE[26]. We observed that IFN-β treatment induces higher levels of IL-6, so we hypothesized that blocking IL-6 would ameliorate disease in TH17-EAE mice treated with IFN-β. To address our hypothesis, TH17-EAE mice were treated with IFN-β or vehicle as well as with an antagonistic anti-IL-6R antibody or isotype control. We found that treatment with anti-IL-6R did not ameliorate TH17-EAE in vehicle-treated mice (Fig. 7a). In agreement with the

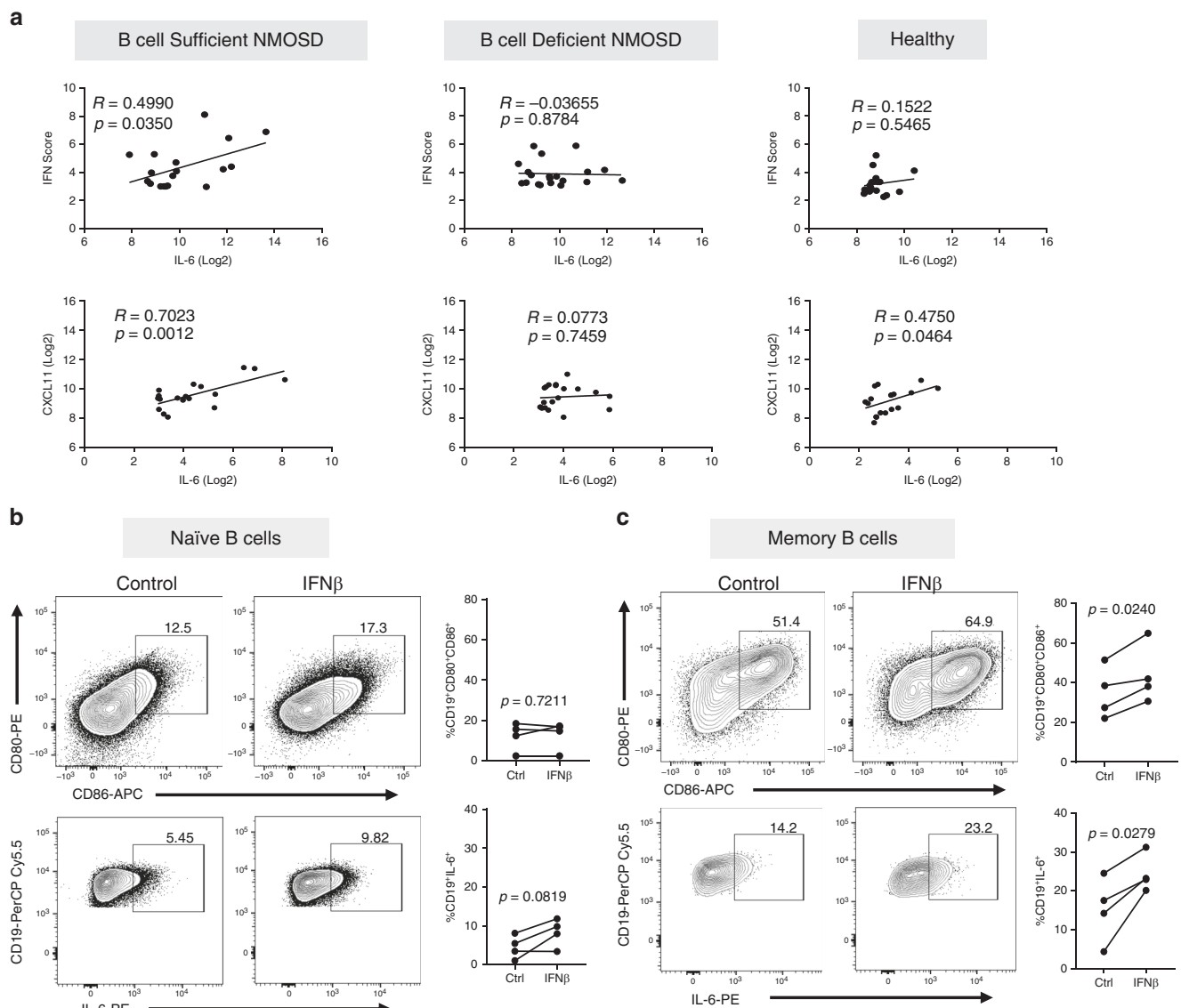

**Fig. 5 Type I IFN drives IL-6 production in human memory B cells. a** Correlation between composite IFN scores and serum CXCL11 levels with IL-6 in B cell-sufficient NMOSD patients ($n = 18$), B-cell-deficient NMOSD patients ($n = 20$) and healthy volunteers ($n = 18$). R and p values were determined using two-tailed Pearson correlation coefficient tests. Direct effect of IFN-β on human naive and memory B cells was assessed by stimulating purified human naïve (CD27−) and memory B cells (CD27+) from healthy donors ($n = 4$) with CD40L, anti-Ig ± IFN-β. Representative flow cytometric plots and frequency of live **b** naive CD19+ B cells and **c** memory B cells expressing CD80, CD86, and IL-6 are shown. Statistical significance was determined using two-tailed Student t tests. P values <0.05 were considered significant.

clinical course of vehicle-treated mice, we found no significant difference in the number of T helper cells secreting GM-CSF or IL-17 (Fig. 7b) and no difference in the number of neutrophils or inflammatory monocytes (Supplementary Fig. 3a) in the spinal cords of vehicle-treated mice. This result is similar to previous reports showing that inhibition of IL-6 does not effectively reduce adoptive transfer EAE[26,27]. In contrast, we found that treatment with both IFN-β and anti-IL-6R significantly attenuated TH17-induced EAE (Fig. 7c). Reduced numbers of GM-CSF+ and IL-17+ TH cells (Fig. 7d) and neutrophils (Supplementary Fig. 3b) was observed in the CNS of mice treated with both IFN-β and anti-IL-6R. These data thus show that IFN-I drives an inflammatory function of IL-6 in TH17-EAE.

**B-cell-deficiency reduces TH17-EAE regardless of IFN-I.** Several studies support the importance of B cells in driving TH17-

induced neuroinflammation[28–30]. However, how Type I IFN affects the function of B cells during TH17-EAE is not known. To address this question, we induced TH17-EAE in C57BL/6 mice and in B-cell-deficient, μMT mice and then treated with either vehicle or IFN-β. We found that vehicle-treated μMT mice had a significant delay in the onset of TH17-EAE disease as compared to vehicle-treated C57BL/6 mice (Fig. 8a). However, we found that at the experimental endpoint, μMT mice had similar disease scores to C57BL/6 mice. In fact, at disease endpoint, we found no significant difference in the number of T helper cells secreting GM-CSF or IL-17 in the spinal cords of vehicle-treated μMT and C57BL/6 mice (Fig. 8b). IFN-β-treated μMT mice had significantly attenuated disease severity throughout the entire course of disease in comparison with C57BL/6 mice (Fig. 8c). In agreement with the disease course of IFN-β-treated mice, there was a significant reduction in the number of T helper cells expressing GM-CSF or IL-17 in the spinal cords of μMT mice (Fig. 8d). Together, these findings show

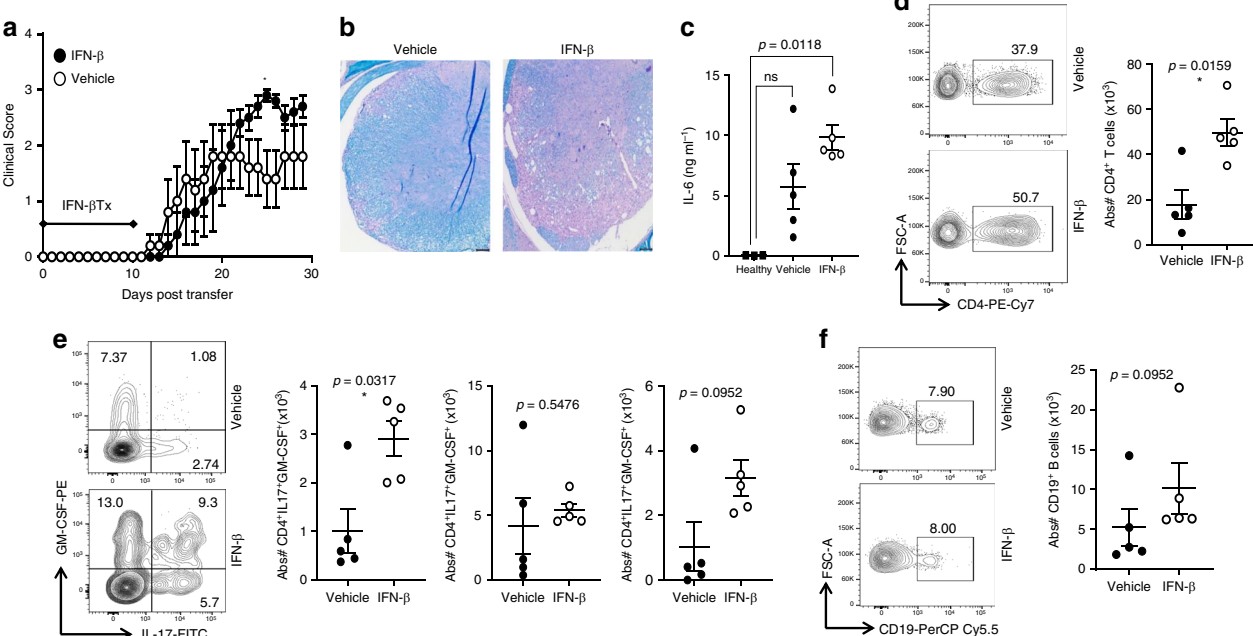

**Fig. 6 IFN-β elevates IL-6 and exacerbates TH17-driven EAE.** MOG-primed TH17 cells were transferred into recipient mice and treated with 1000 U of IFN-β or vehicle every other day from day 0–10 post transfer of cells. **a** Clinical scores of mice with TH17-EAE treated with vehicle or IFN-β; (TH17-vehicle $n = 5$, TH17-IFN-β $n = 5$) and two-tailed Mann–Whitney tests were performed to determine statistical significance (*$P < 0.05$). Representative of four experiments with similar results. **b** Spinal cord sections from mice (day 30) were stained with H&E and Luxol fast blue. Representative images of four mice in each group. Scale bar represents 100 μm. **c** Levels of IL-6 in the sera (measured by ELISA) of TH17-EAE (day 2) were elevated with IFN-β; (TH17-vehicle $n = 5$, TH17-IFN-β $n = 5$, healthy $n = 3$) and a two-tailed Kruskal–Wallis test with multiple comparisons corrected by the Dunn's method was used to determine statistical significance. **d** Representative flow cytometry plots and absolute numbers of live CD4+ T cells in the spinal cords of EAE mice, 30 days post transfer. **e** Representative flow cytometry plots and absolute numbers of CD4+ T cells expressing IL-17, GM-CSF or both in the spinal cords of EAE mice. **f** Representative flow cytometry plots and absolute numbers of live CD19+ B cells in the spinal cords of EAE mice. (TH17-vehicle $n = 5$, TH17-IFN-β $n = 5$) and two-tailed Mann–Whitney tests were performed to determine statistical significance. Error bars indicate the S.E.M. Source data are provided as a Source Data file.

that B cell-deficiency reduces TH17-EAE disease severity, regardless of IFN-I treatment. Furthermore, our data also show that protection of disease by B cell-deficiency is prolonged in IFN-I-treated mice compared with vehicle-treated mice.

**IFN-β stimulates B cells to drive pathogenic TH17 cells**. Our data demonstrate a link between IFN-I, B cells, and IL-6 to elevated inflammatory TH17 responses in neuro-inflammation. These observations led us to hypothesize that B cells are the inflammatory mediator between IFN-I and TH17 during inflammation. To test this hypothesis, we designed the following culture experiment. We first stimulated B cells isolated from healthy or EAE mice in the presence or absence of IFN-β, washed the B cells of IFN-β, then co-cultured these B cells with CD4+ T cells from 2D2 mice[31] in the presence of the myelin peptide antigen, MOG$_{35-55}$ (Supplementary Fig. 4a).

Prior to treatment with IFN-β, we observed that B cells isolated from EAE mice had a more mature phenotype (IgM$^{hi}$IgD$^{hi}$ and IgM$^{lo}$IgD$^{hi}$) compared with B cells from healthy mice (Supplementary Fig. 4b). We also evaluated the expression of IFN-αβ receptor (IFNAR) and found no difference in IFNAR expression in B cells from healthy and EAE mice (Supplementary Fig. 4c). We found that IFN-β directly increased the expression of CD80, CD86, and MHCII on B cells isolated from both healthy and EAE mice (Fig. 9a, b, Supplementary Fig. 4d), suggesting that IFN-β enhances the antigen-presenting function of B cells. IFN-β stimulation of B cells isolated from healthy mice had marginal effects on IL-6, IL-12/IL-23p40, and IL-10 secretion (Fig. 9c).

Strikingly, IFN-β stimulation of B cells from EAE mice led to an abundance in secretion of IL-6 and IL-12/IL-23p40 but not IL-10 (Fig. 9c). These data suggest that IFN-β has a direct effect on a population of mature B cell that results in its skewing towards a more inflammatory phenotype.

We next assessed for 2D2 T-helper cell proliferation and cytokine production from the co-culture assay. We observed that there was no significant effect on T-cell proliferation following co-culture with IFN-β-stimulated B cells from healthy mice (Fig. 9d, Supplementary Fig. 4e). In contrast, IFN-β stimulation of EAE-derived B cells significantly increased T-cell proliferation (Fig. 9e, Supplementary Fig. 4e). In addition, IFN-β-stimulated B cells from healthy mice did not impact the secretion of IL-17, GM-CSF and IL-10 by T helper cells (Fig. 9f). However, we found enhanced secretion of GM-CSF and IL-17, but not IL-10, from T helper cells co-cultured with IFN-β-stimulated B cells isolated from EAE mice (Fig. 9f). These cell culture assays demonstrate that IFN-β acts directly on antigen-experienced B cells to elevate their expression of CD80, CD86, MHCII, IL-6, and IL-12/IL-23 p40, which in turn drive the proliferation of inflammatory T helper cells that secrete elevated levels of IL-17 and GM-CSF.

## Discussion

The complex interplay between IFN-I and TH17 cells plays a significant role in the pathology of certain autoimmune diseases, notably, MS, NMOSD, psoriasis, systemic lupus erythematosus and ulcerative colitis[32]. The ability of IFN-I to drive or inhibit inflammation relies on the disease context. IFN-β remains a

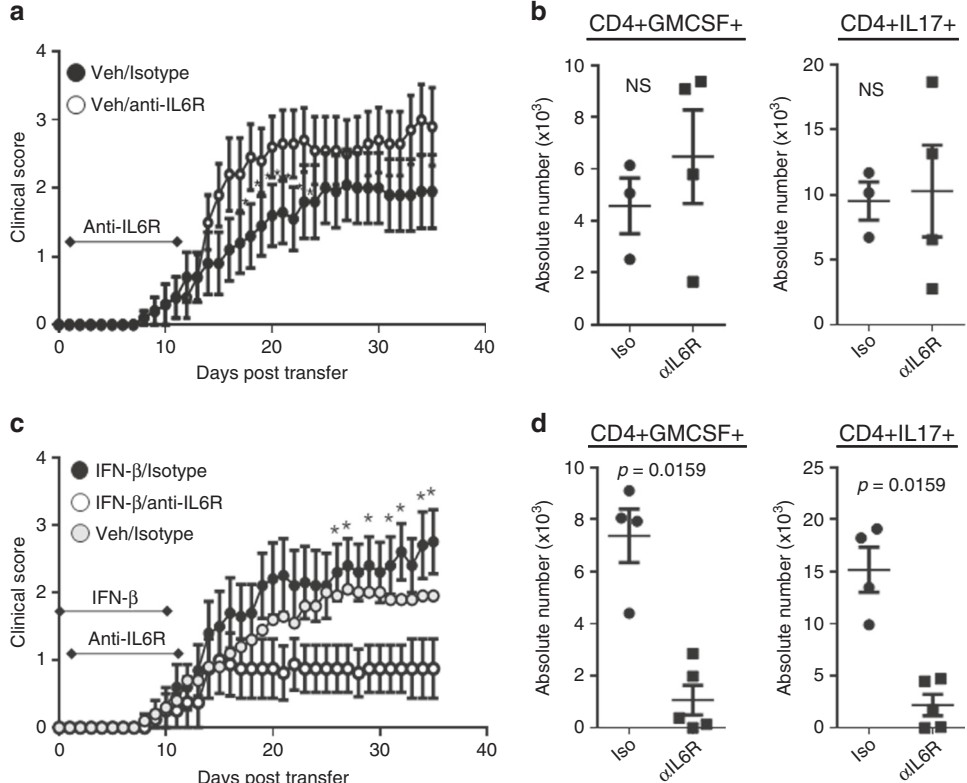

**Fig. 7 IL-6 blockade attenuates IFN-β-treated TH17-EAE.** TH17-EAE mice, either treated with IFN-β or vehicle, were also treated with either anti-IL-6R or an isotype control every 5 days from days 1–11. **a** Effect of treatment with anti-IL-6R (n = 10) or isotype (n = 10) control on the clinical scores of vehicle-treated TH17-EAE. Data were pooled from two independent experiments. **b** Number of CD4+ T cells that express GM-CSF and IL-17 in spinal cords of vehicle-treated mice (day 35). **c** Effect of anti-IL-6R (n = 8) or isotype (n = 10) treatment on the clinical scores of IFN-β-treated TH17-EAE. Data were pooled from two independent experiments. Vehicle/isotype treated mice were also plotted for reference. **d** Number of CD4+ T cells that express GM-CSF and IL-17 in the spinal cords of IFN-β-treated EAE mice (day 35). Statistical analysis was performed using Mann–Whitney tests (P < 0.05). Error bars indicate the S.E.M. Results are compiled from two independent experiments.

widely prescribed treatment for MS. As a therapy, IFN-I reduces relapse rates and lesion formation in MS patients and a predominant theory behind its efficacy is through the inhibition of TH17 differentiation and function[17]. Paradoxically, strong evidence from NMOSD and TH17-EAE indicate that IFN-I and TH17 cells cooperate to drive disease progression[10,33–35]. Our study now defines a mechanism by which IFN-I cooperates with TH17 to drive severe disease in NMOSD.

We now show that IFN-I signatures stratify NMOSD patients into two subsets: IFN-high & IFN-low. Our data indicate that IFN-high NMOSD patients have elevations in IL-6 and cytokines related to the TH17 pathway. Most strikingly, IFN-I signatures and serum IL-6 stratify patients based on disability highlighting their potential utility in clinical tests for the prognosis of NMOSD. In addition, we found that patients treated with rituximab had reduced IL-6 and IL-17 levels in IFN-high NMOSD patients. Currently, the precise mechanisms through which rituximab mediates its therapeutic effects is unclear, but these data suggest that the therapeutic mechanism is through the reduction of IL-6. This observation is congruent with previous reports showing that reducing IL-6 expressing B cells is critical for the therapeutic effects of rituximab in mice with EAE[36]. Therefore, we speculate that IL-6 levels could be used to monitor treatment response to B cell-depleting therapies in NMOSD. Our data also indicate that IFN-I stimulation is responsible for the elevated IL-6 production from memory B cells. In NMOSD, we found a significant correlation between IFN-I signature

expression and IL-6 levels, and this correlation is absent in patients treated with rituximab. B cell cultures also determined that the human memory B cell population produces high levels of IL-6 after IFN-I stimulation. In summary, these data suggest an inflammatory cascade that is initiated by IFN-I to induce IL-6 from memory B cells, which then affects other inflammatory pathways, such as, the generation of inflammatory TH17 cells and autoantibody production.

Classification of MOG seropositive patients in the NMO spectrum is currently being re-evaluated[37]. In our cohort, we find that both AQP4-IgG+ and MOG-IgG+ patients have elevated IFN-I signatures compared to healthy controls. This suggests that IFN-I pathway drives disease in both MOG-IgG+ and AQP4-IgG+ patients and perhaps therapeutic strategies that block IFN-I would be effective in both patient subsets. Recent studies have identified that MOG-IgG+ NMOSD patients do not respond equally well to B cell depletion compared to AQP4-IgG+ patients[23]. In addition, another study suggested that IL-6R inhibition might be more effective in AQP4-IgG+ patients than in AQP4-IgG− patients[38]. Our data revealed that relapse rates of B cell depleted MOG-IgG+ patients were significantly higher than relapse rates of B cell depleted AQP4-IgG+ patients. We did not observe differences in the transcriptomes or serum proteins in B cell depleted AQP4-IgG+ or MOG-IgG+ patients. Interestingly, in B cell-sufficient patients, we found elevated serum levels of IL-6 and the IFN-I chemokine, MCP-3 in AQP4-IgG+ but not MOG-IgG+ patients. Although longitudinal studies are needed, we

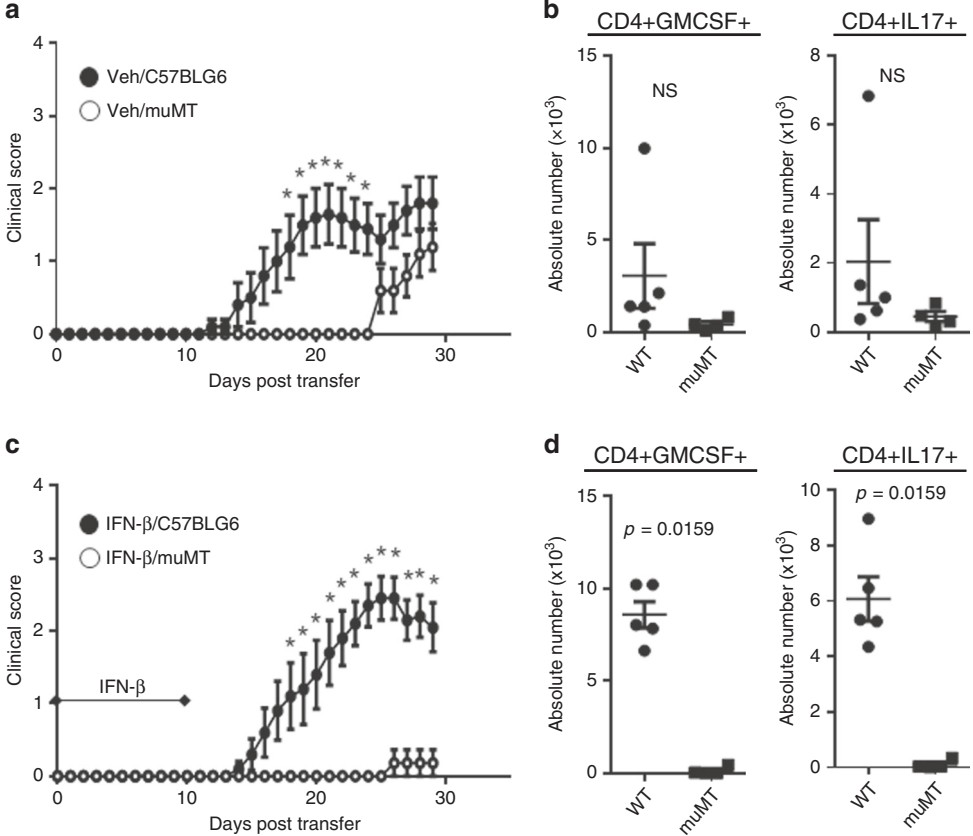

**Fig. 8 B cell-deficiency attenuates IFN-β-treated TH17-EAE.** TH17-EAE was induced in either C57BL/6 or μMT mice and treated with IFN-β or vehicle. **a** Clinical scores of vehicle-treated C57BL/6 ($n = 10$) and μMT ($n = 10$) mice with TH17-EAE. Data were pooled from two independent experiments. Mann–Whitney tests were performed to determine statistical significance ($P < 0.05$). **b** Number of CD4+ T cells that express GM-CSF and IL-17 in spinal cords of vehicle-treated mice (day 29). **c** Clinical scores of IFN-β-treated C57BL/6 ($n = 10$) and μMT ($n = 11$) mice with TH17-EAE. Data were pooled from two independent experiments. Mann–Whitney tests were performed to determine statistical significance ($P < 0.05$). **d** Number of CD4+ T cells that express GM-CSF and IL-17 in spinal cords of IFN-β-treated mice (day 29). Statistical analysis was performed using two-tailed Mann–Whitney tests. Error bars indicate the S.E.M. Results are compiled from two independent experiments. Source data are provided as a Source Data file.

speculate that IFN-I and IL-6 pathways are involved in the responsiveness to B cell depletion and IL-6R inhibition in AQP4-IgG+ NMOSD patients.

The TH17-EAE model in C57BL/6 mice mimics several aspects of NMOSD[9,13], which demonstrates the usefulness of this animal model for mechanistic studies of NMOSD. Here, we found that the results from our TH17-EAE experiments are congruent with the observations made with the NMOSD patient specimens. We found that creating an IFN-I-high TH17-EAE model, with IFN-β injections, resulted in increased serum IL-6, elevated TH17 responses and exacerbated paralysis in mice. Clinical trials demonstrate that IL-6R inhibition and B cell depletion are promising therapies for NMOSD[39]. To address how IL-6 blockade would affect IFN-β treatment of TH17-induced disease, TH17-EAE mice were treated in vivo with or without IFN-β as well as with anti-IL-6R or an isotype control. Surprisingly, IL-6 blockade in TH17-EAE without IFN-β treatment did not ameliorate disease. A possible explanation for this observation is that IL-6 is required for the generation of inflammatory TH17 cells and blockade is no longer effective in the adoptive transfer model of EAE where TH17 are already activated. In striking contrast, IL-6 blockade significantly attenuated TH17-EAE treated with IFN-β, demonstrating that IL-6 is a critical inflammatory mediator induced by IFN-I which exacerbates disease. Contrary to IL-6R inhibition of TH17-EAE, we found that B cell-deficiency

attenuates disease in TH17-EAE regardless of IFN-β treatment. These data suggest that B cells play a key role in initiating disease in TH17-EAE mice, which is not mediated through a IFN-I/IL-6 cascade but likely through antigen presentation[40,41]. The differences in efficacy of IL-6 inhibition and B cell-deficiency in IFN-β treated TH17-EAE may provide insights into how IFN-high and IFN-low NMOSD patients will respond to these therapies. In this study, we use the adoptive transfer of MOG-specific TH17 cells to induce disease that reflects many aspects of NMOSD, especially in regards to IFN-I. The development of a mouse model with a neuro-autoimmune disease that targets AQP4 has been a challenge. Recent developments have shown that T cells from AQP4-deficient mice recognize distinct AQP4 epitopes and these AQP4-specific T cells require TH17 programming to induce severe optico-spinal inflammation[42]. These data confirm the importance of the TH17 pathway in driving NMOSD-like disease in mice. However, how IFN-I affects the AQP4-specific animal model remains to be tested.

One predominant theory behind the efficacy of IFN-β is that this therapy reduces disease by inhibiting TH17 differentiation and function[16–18]. However, MS and NMOSD patients with high TH17 signatures and mice with TH17-induced EAE have exacerbated disease when treated with IFN-β[4,5,8,10,11,14,43]. Our cell culture experiments provide key insights into how IFN-β paradoxically increases TH17 pathology (depicted in Fig. 10). IFN-β indirectly

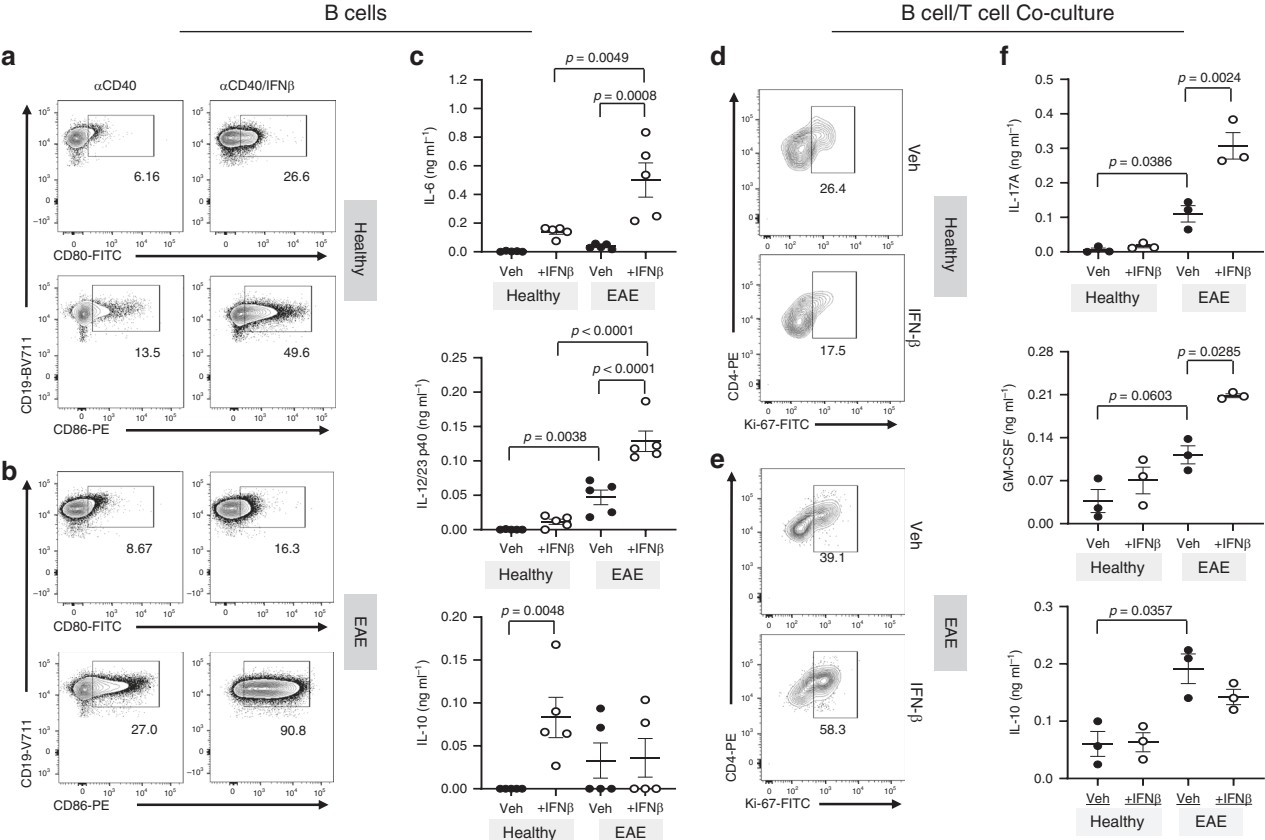

**Fig. 9 Effect of IFN-β-stimulated B cells on TH17 cells.** Purified B cells from spleens of healthy or EAE mice ($n = 5$) were stimulated with anti-CD40 ± IFN-β for 3 days. Following stimulation, B cells were washed and co-cultured with CD4$^+$ T cells from 2D2 mice in the presence of MOG$_{35-55}$ antigen. Following IFN-β stimulation, B-cell phenotype and cytokine production were assessed by **a**, **b** FACS and **c** ELISA, respectively. Stimulated B cells were washed and co-cultured with antigen-specific 2D2 T-helper cells in the presence of MOG$_{35-55}$ antigen. Representative flow cytometry plots of Ki-67 staining of 2D2 CD4$^+$ T cells cultured with B cells from **d** healthy ($n = 3$) or **e** EAE mice ($n = 3$). **f** The cytokines IL-17A, GM-CSF, and IL-10 from the co-culture supernatants were analyzed by ELISA ($n = 3$ per group). Statistical significance was determined using paired one-way ANOVA tests with multiple comparison corrections using the Holm-Sidak's method. $P$ values < 0.05 were considered significant. Error bars indicate SEM. Source data are provided as a Source Data file.

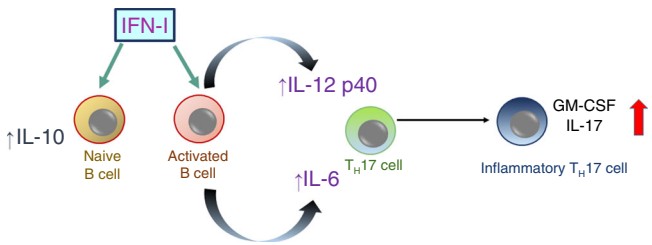

**Fig. 10 IFN-I indirectly promotes TH17 pathogenicity.** Data from Fig. 9 indicate that IFN-I stimulates the expression of IL-6 and IL-12p40 from activated B cells which, in the context of auto-antigen, supports the proliferation of inflammatory TH17 cells. In contrast, IFN-I stimulation of naive B cells elevates IL-10 and not IL-6 and does not efficiently promote inflammatory TH17 cell proliferation.

enhances the pathological functions of TH17 cells by increasing IL-6 and IL-12p40 (IL-23) secretion from activated/memory B cells. Since auto-reactive B cells are integral in NMOSD and other autoimmune diseases, it would be of interest to determine if IFN-I would have different effects on self-reactive B cells and foreign-reactive B cells in diseased and healthy individuals.

Altogether, this study provides novel insights into how IFN-I drives pathology in diseases with elevated TH17 signatures, such as NMOSD. Our data from patient sera and mice with TH17-EAE indicate that IFN-I induces IL-6 to drive TH17 neuro-

inflammation. Our observations suggest that IFN-I alters both the transcriptional and cytokine profiles towards an inflammatory phenotype during TH17-mediated disease. IFN-I-driven TH17 pathogenicity occurs in an indirect manner and is partly attributed to the effect of IFN-I on B cells. Overall, these findings broaden our understanding of what biological pathways drive severe disease in NMOSD and provide potential markers for the clinical management of these patients. Further studies from larger cohorts are underway to confirm the clinical relevance of IFN-I and TH17 biomarkers in this devastating neuro-inflammatory disease.

## Methods

**Clinical classification of NMOSD.** We obtained serum and PAXGene tubes from 42 patients with NMOSD from the Charité-Universitätsmedizin Berlin (Supplementary Table 1). Serum proteins were measured in all 42 patients. Thirty-eight of the 42 NMOSD RNA samples passed quality control and were analyzed by RNAseq. Of the 42 patients, EDSS was available for 40 patients and relapse rates were available for 41 patients. We obtained PBMCs from seven NMOSD patients, six of which had EDSS scores, from the University of Michigan (Supplementary Table 2). NMOSD diagnosis was fulfilled using clinical criteria defined by Wingerchuk et al.[44]. All NMOSD patients were tested for AQP4-IgG or MOG-IgG using cell-based assays (CBA)[45,46]. No patient was on steroid therapy during blood draw. Serum and PAXgene tubes were obtained from 18 healthy volunteers, 18 passed quality control and were used in transcriptomic and proteomic analysis. PBMCs were obtained from 13 healthy volunteers and used for FACs analysis. Written informed consent was obtained from individuals prior to participation in the study, which was approved by the Charité Universitätsmedizin Berlin, University of Michigan and the Oklahoma Medical Research Foundation's Institutional Review Boards. Assays on the samples were performed blinded from the clinical data.

**Peripheral blood RNA transcript isolation**. Whole blood was obtained by venipuncture into PAXGene tubes (BD company) and RNA was extracted with on-column DNase digestion (Qiagen). Excess globin transcripts were removed using GLOBINclear (Ambion). RNA concentrations were determined using a NanoDrop spectrophotometer and RNA quality was assessed using the RNA 6000 Nano kit on the Bioanalyzer 2100 (Agilent) with quality threshold RIN scores > 8.

**RNASequencing and quality control measures**. Starting from the raw FASTQ files (2 × 100bp), the quality of raw sequence reads was assessed using FASTQC v0.11.5 (https://www.bioinformatics.babraham.ac.uk/projects/fastqc/). The reads were then trimmed using Trimmomatic v0.35[47] to remove low-quality reads. The quality of the reads was then re-assessed using FastQC to confirm quality improvements. All downstream analyses were based on the clean data with the highest quality. The raw FASTQ files are aligned to the human reference genome (GRCh38) using HISAT2 v2.0.4[48] and the aligned files were sorted to bam files using SAMTOOLS v1.9[49]. $1.5–2.0 \times 10^8$ mapped reads were obtained per sample. The sequencing performance was assessed for total number of mapped reads, total number of uniquely mapped reads, strandedness, genes, and transcripts detected, ribosomal fraction known junction saturation, and reads distribution over known gene models with RSeQC v3.0.0[50]. Sample quality control was assessed using ArrayQualityMetrics v3.14.0[51] in R. Out of a total of 61 samples, five samples were considered poor-quality and removed from subsequent analyses based on: (a) deviation of read counts, assigned to features from mean ± 2 SD of all samples, (b) having strandedness issue detected by RSeQC, and (c) being detected as outliers by ArrayQualityMetrics. Therefore, RNASeq data from 38 patients with NMOSD and 18 healthy controls were used for subsequent analyses. Transcript counts were derived from the uniquely aligned unambiguous, strand-specific (reverse-stranded) reads by Subread:featureCount v1.6.3[52], yielding 58,052 transcripts per sample. To assess cell type-based RNA expression, we used a Genome-wide RNA database (www.proteinatlas.org). IFN scores were calculated as a log2 average read count of the 25 IFN genes identified as elevated in NMOSD (Fig. 1d).

**Serum protein profiling of NMOSD and RRMS patients**. Protein arrays were performed on sera drawn during stable disease from 42 NMOSD patients and 18 healthy volunteers. Concentrations of 91 proteins were assessed by proximity extension assay (Olink Bioscience, Sweden) using the Inflammation panel. The assay uses oligonucleotide-labeled antibody pairs allowing for pair-wise binding to target proteins. When antibody pairs bind target antigens, corresponding oligonucleotides form an amplicon allowing for quantification of protein expression by high-throughput real-time PCR. Data are presented as normalized protein expression values, Olink Proteomics' arbitrary unit on a log2 scale.

**Flow cytometric analysis of T helper cells in NMOSD patients**. Heparinized blood was collected from each patient in BD VacutainerTM Sodium HeparinN green top tubes. The green top tubes were mixed and centrifuged for isolation of PBMCs at the cell layer. Plasma was then removed, aliquoted and stored at −80 °C. PBMCs were stained with antibodies to mark T-helper cell subsets and analyzed by flow cytometry using the BD FACSCantoII system and FlowJo. TH1 cells were defined as CD4+ CXCR3+CCR6−CD161−; TH17.1 were defined as CD4+CXCR3+CCR6+CD161+ and TH17 cells were defined as CD4+CXCR3−CCR6+CD161+ (Supplementary Fig. 2).

**In vitro IFN-β stimulation of human B cells**. Fresh PBMCs from four healthy donors were isolated using Ficoll-Paque Plus (GE Life Sciences). Memory and naïve B cells were purified from PBMCs with human anti-CD27 conjugated beads (Miltenyi). $0.275 \times 10^6$ cells/ml of each B cell subset was stimulated with anti-human IgM/IgG (3 µg/ml Jackson Immunoresearch) and CD40L (1 µg/ml, R&D), with or without IFN-β (1000U/ml, PBL) for 3 days. Human B cells were Fc blocked (BD Biosciences), stained with viability dye (Tonbo) and fluorochrome-conjugated anti-human mAbs: CD19 (Biolegend), CD86 (Biolegend) and CD80 (Biolegend). For cytokine staining, B cells were stimulated with PMA (Sigma-Aldrich), ionomycin (Sigma-Aldrich) and Brefeldin A (GolgiPlug, BD Bioscience) for 4 hours. Cells were stained with viability dye (Tonbo), CD19 (eBioscience), and IL-6 (Biolegend).

**Mice**. Eight to ten-week-old female C57BL/6, µMT, and 2D2 Tg mice were purchased from Jackson Laboratory, housed in the Oklahoma Medical Research Foundation animal facility and treated in compliance with the institutional IACUC.

**TH17-EAE induction and treatment**. Donor C57BL/6 mice were immunized with 150 µg MOG$_{35–55}$ peptide (Genemed Synthesis Inc.) emulsified in CFA (5 mg/ml heat-killed Mycobacterium tuberculosis), subcutaneously. This was followed by an intraperitoneal (I.P.) injection of 250 ng of Bordetella pertussis toxin (Ptx) (List Biological Laboratories Inc.) in PBS on day 0 and day 2 post immunization. Donor mice were killed on day 10 post-immunization. Spleens and lymph nodes were harvested, mechanically disrupted to obtain a single-cell suspension and $2.5 \times 10^6$ cells/ml were stimulated for 3 days with MOG$_{35–55}$ (10 µg/ml), IL-23 (10 ng/ml; R&D Systems), and anti-IFN-γ (10 µg/ml; eBioscience) in complete RPMI 1640

(Gibco). C57BL/6 or µMT recipient mice were I.P. injected with $15 \times 10^6$ cells and treated with IFN-β (1000 U/ml; PBL) or PBS on days 0, 2, 4, 6, 8, and 10. Recipient mice also received Ptx on day 0 and day 2 post transfer. Mice were monitored daily for clinical scores. Paralysis was assessed using the following standard clinical score: (0) healthy, (1) loss of tail tone, (2) partial hind-limb paralysis, (3) complete hind-limb paralysis, (4) forelimb paralysis, and (5) moribund/dead. Transfer EAE mice were killed on day 15 and spinal cords were fixed and sectioned for histological analysis using H&E and Luxol fast blue staining. Serum was collected on day 2 post transfer and IL-6 expression was assessed with an anti-mouse IL-6 ELISA kit (eBioscience). For in vivo IL-6 blockade, mice were treated with an anti-IL-6R antibody or IgG2b isotype control (8 mg/dose; BioXCell) on days 1, 6, and 11 post transfer. Treatments were carried out in a blinded experiment. At disease endpoint, CNS infiltration by immune cells was assessed by perfusing EAE mice with PBS and collecting their brains and spinal cords. CNS tissue was homogenized through mechanical disruption and homogenates were incubated with DNAse (5 µl/ml; Sigma) and collagenase (4 mg/ml; Roche) at 37 °C for 1 hour. Cells were isolated using a Percoll gradient and analyzed by FACS.

**Flow cytometry of mouse cells**. All cells were stained with Fixable Viability dye (eBioscience, Biolegend) and treated with Fc block (eBioscience) prior to staining with fluorochrome-conjugated anti-mouse mAbs. mAbs were from Biolegend (F4/80, Ly6C, Ly6G, IgD, CD19, CD86, CD80, IFNAR, IgG1) and eBioscience (CD11b, MHCII, IgM).

For intracellular cytokine staining, cells were stimulated with PMA (Sigma-Aldrich), ionomycin (Sigma-Aldrich) and monensin (BD Biosciences) for 4 hours. Cells were then stained with anti-mouse CD4 (eBioscience), fixed and permeabilized with Cytofix/Cytoperm (BD Biosciences) and stained for IL-17 (BioLegend) and GM-CSF (Biolegend). All flow cytometric data were collected on LSRII (BD Biosciences) and analyzed using FlowJo software (Tree Star Inc.).

For intranuclear staining of Ki-67, cells were stained with anti-mouse CD4 (BDBioscience), fixed and permeabilized using the Foxp3 Transcription Factor Staining Buffer Set and stained for Ki-67 (Biolegend) to assess for cellular proliferation.

**Mouse B cell and T-cell co-culture assays**. Spleens from either healthy or EAE mice (10 days after immunization with MOG$_{35–55}$/CFA and PTX) were harvested and processed. Purified B cells from splenocytes were obtained using negative sorting with magnetic beads (Miltenyi). Isolated B cells ($2.5 \times 10^6$ cells/ml) were stimulated with anti-CD40 (1 µg/ml, eBioscience) with or without IFN-β (100 U/ml) for 3 days. B cells from healthy or EAE mice stimulated with or without IFN-β were then washed with PBS (1×) and co-cultured with magnetically sorted TH cells (Miltenyi) from 2D2 mice. B and TH cells were co-cultured ($2.5 \times 10^6$ cells/ml) at a 1:1 ratio with MOG$_{35–55}$ antigen.

**Statistical analysis**. For RNA Sequencing, genes with less than one count per million for at least in 1/3rd of samples were considered as non-expressed and not used for differential expression analyses. This resulted in 40,796 transcripts being removed out of a total of 58,052 transcripts, leaving 17,256 for further analysis. Differential gene expression analyses were performed using DESeq2 v1.24[53], fitting a negative binomial generalized linear model to find significantly DEGs. Genes with a false discovery rate of 0.05 and fold change ≥0.58 or ≤0.57 were considered differentially expressed. All analyses were performed in the R Bioconductor suite.

Data were measurements from distinct biological replicate samples and are presented as means ± s.e.m. and statistical significance was determined using two-tailed Student's t tests or Mann–Whitney tests. In the case of three or more data sets, means were compared using two-way analysis of variance with Bonferroni correction or Kruskal–Wallis with a Dunn's multiple comparison test. Differences were considered significant for $P < 0.05$. Statistical analyses were performed using Prism 6 (GraphPad). All statistical tests were two-tailed. Cluster analysis of NMOSD patients was performed using hierarchical clustering with Gene Cluster software, where the log2 cytokine values were centered to the mean, then ordered by complete linkage clustering[54]. The clusters were presented as a heat map using TreeView[54].

**Reporting summary**. Further information on research design is available in the Nature Research Reporting Summary linked to this article.

## Data availability

The consent form signed by the participants in this study does not permit public release of potentially identifiable data, which includes the deposit of raw RNAsequencing data. We have provided the read counts in source data file and the raw RNA-sequence data are available from the authors. All other data are provided in the source data file. Source data are provided with this paper.

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

## Acknowledgements

This manuscript was funded by grants awarded to Dr. Axtell from the National Multiple Sclerosis Society (RG-1602-07722), the National Institutes of Health (R01AI137047 and R01EY027346) and to Dr. Paul and Dr. Ruprecht from the BMBF Competence Network Multiple Sclerosis and Dr. Paul from the German Research Council DFG.

## Author contributions

A.A., F.P., and R.C.A. conceptualized, designed experiments, interpreted results, and wrote the manuscript. A.A., S.G., G.K., J.L.Q., R.M.K., and R.C.A. executed and analyzed the animal experiments. N.B., K.R., F.P., Y.M.-D. established the patient cohorts. A.A., N.B., S.G., F.P., and R.C.A. analyzed the patient serum data with clinical data. Q.W. analyzed the T-cell subsets in PBMCs from patients. C.J.L. and B.K. performed Q.C. and analysis of the RNAseq data.

## Competing interests

R.C.A. has consulted for Roche, Biogen, and EMD serono. Y.M.-D. has consulted for and/or received grant support from: Acorda, Bayer Pharmaceutical, Biogen Idec, EMD Serono, Genzyme, Novartis, Questor, Genentech, and Teva Neuroscience. F.P. has consulted for and/or received speaker honoraria from Bayer, Teva, Genzyme, Merck, Novartis, and MedImmune. All other authors declare no competing interests.
