## [Peer Review File · Nature Communications]

Reviewers' comments:

Reviewer #1 (Remarks to the Author):

The research by Agasing et al elucidates the biological cascades that involves type 1 interferon (IFN-I), IL-6, B cells and TH17 cells in neuromyelitis optica spectrum disorder (NMOSD) and in TH17-induced experimental autoimmune encephalomyelitis (TH17 EAE). In this study, from the proteomics and transcriptomic profiles, disability in patients was associated with increased levels of IL-6, IL-17 and IFN-I. In patients, B cell depletion therapy reduced the levels of IL-17 and IL-6 in a group defined as expressing high levels of IFN-I genes. In TH17-induced passive EAE, mice treated with IFN-I increased the levels of IL-6 and worsened disease. Interestingly blocking of IL-6 only reduced disease in TH17 EAE mice treated with IFN-I. On the other hand, the disease was attenuated in both IFN-treated and untreated mice when B-cells were absent. The paper describes a model in which IFN-I drives B production of IL-6 which results in Th17 pathogenicity. The summary of the paper contradicts a preexisting paradigm that IFN-I inhibits TH17 function. It is beneficial to understand how the effects of IL-6 inhibition and B cell depletion therapy vary between IFN-I high and low NMOSD patients. The paper also has potential implications for prognostic evaluation measures in NMOSD.

This is an extensive study that has a tremendous amount of information related to the pathogenesis of NMOSD and neuroinflammation. The exhaustive phenotyping of the human subjects alone is quite noteworthy. The authors should also be recognized for the expert blend of human and animal modeling as well as the apt use of in vitro cell culture experiments to formulate a model that has a significant bearing on the understanding of NMOSD. In fact, the statement by the authors that this work addresses "a major knowledge gap in the field of neurology" should be extended to immunology and the fields of translational neuroscience.

Three concerns - that if addressed would improve the delivery of the study's message - are as follows:

1. A visual model incorporated into the figures would be highly useful. In particular, the role of different B cell subsets, the presence of B cells in the periphery vs. the CNS, and the signaling between T cells and B cells would be more clearly conveyed in a visual manner.
2. Additional information from a few experiments would be useful for more definitive conclusions. In particular, in Figure 5c a control without any interferon or anti-IL-6R in parallel would allow for the conclusion that the combination is suppressive of disease, rather than to a severity exhibited by non-interferon treated mice. Also, in the model incorporating B cells receptiveness to IFN-I in order to facilitate T cells response, expression of IFNAR by B cells would be important. Further, addition of anti-IFNAR antibodies in the experiments shown in Figure 6C & Figure 6D would greatly aid the conclusion.
3. It would be appropriate to tone down assertions of prognosis in NMOSD based on this study as it does not include longitudinal data.

A few minor changes would also be worthy of addressing:

1. Statistical considerations are important; the number of subjects included is impressive given the prevalence of NMOSD. However, how definitive are the divisions of IFN-I categories if the number of subjects is so low? An expanded discussion regarding how the findings will hold up amongst a larger cohort would help.
2. The assertion that anti-MOG and anti-AQP4 NMOSD disease states are founded on a common pathogenesis based on this study is an overstatement. This should be emphasized less.
3. The antigen specificity of B cells should be discussed, particularly as the underpinnings of NMOSD appears to be antigen driven. How antigen-specific B cells may exhibit particular responsiveness to

IFN-I may be critical to the model.

Reviewer #2 (Remarks to the Author):

In this manuscript, Agasing, et al. examined the role of the type I IFN pathway in Th17 differentiation in patients with NMO, and in mechanistic studies in mice. Type I IFN, which is used as a treatment in MS, exacerbates NMO, considered a Th17-mediated disease. By transcriptomics (RNA-Seq) on whole blood from NMO patients (treated with rituximab (appx two thirds of NMO patients) or other therapies) and healthy controls (HC) they identified differentially expressed genes between NMO and HC, regardless of rituximab treatment. 26 of the differentially expressed genes were IFN-I-inducible. There were two distinct subsets: IFN-high and IFN-low. By proteomic analysis, IL-6 was one of the most highly upregulated protein in NMO patients. IFN-high NMO patients appeared to have higher EDSS scores. They did not detect a statistically significant association between Th17 and EDSS, although there was a trend. The percentages of IFN-high and IFN-low patients were similar in B cell-depleted (rituximab-treated) patients and patients with B cells (not on rituximab). However, IL-6 and IL-17 were elevated in IFN-high B cell-sufficient patients. By in vitro analysis, they found that IFN-I treatment promoted IL-6 production by CD27+ (memory) B cells.

Th17 MOG EAE was used to further investigate mechanism. IFN-b treatment exacerbated clinical and histologic EAE. Although anti-IL-6R did not significantly block Th17 adoptive transfer EAE, given together, anti-IL-6R did block the IFN-b-exacerbated EAE, indicating IFN-b promotes IL-6-mediated Th17 EAE. B cell-deficient mice developed less severe EAE, independent of IFN-b treatment. To test the hypothesis that B cells mediate between IFN-I and Th17 disease, they evaluated the APC function of IFN-I-treated and untreated B cells in vitro. IFN-b treatment promoted B cell expression CD80 and CD86 costimulatory molecules and MHC II, but marginal increase in IL-6. In contrast, IFN-b treatment of B cells from EAE mice led to increased IL-6 (and IL-12/IL-23p40) production. IFN-b treated B cells from EAE mice promoted T cell secretion of proinflammatory IL-17 and GM-CSF in co-cultures. Overall, their results from NMO patients and in mice indicate that IFN-b treatment promotes B cell IL-6 signature that promotes Th17 disease.

The results in this manuscript are quite exciting. They are of interest to researchers and very relevant to clinicians caring for NMO patients. Although it is well recognized that IFN-b exacerbates NMO and that IFN-b should not be administered to patients with NMO, the mechanism responsible for this effect has been puzzling. This manuscript elucidates a key mechanism; namely B cell IL-6 promotion of Th17 cells is likely responsible for the worsening of NMO by IFN-b. I do have some comments that I believe can be addressed without difficulty.

(1) Rituximab/ocrelizumab treated patients were compared to other NMO patients, which included patients that were treated with other therapies or were not on treatment combined. Did the untreated patients as a group differ (e.g. transcriptome) from patients on treatments other than anti-CD20 treatment?

(2) Both AQP4+ and MOG+ patients were included in this study. Nearly 30% were MOG+. Were there any distinguishing features between these groups? This is potentially important as there are recent publications suggesting that patients with MOG antibody-associated disease do not respond to CD20 B cell depletion as well as patients with AQP4+ NMOSD (e.g. Durozard, P., et al. Ann Neurol 2019). Also, anti-IL-6R treatment was effective in AQP4+ patients, but not AQP4- patients (Yamamura, T., et al. NEJM 2019).

(3) With regard to the antigen presenting capability of B cells, for the in vitro studies shown in Fig 6, the authors focused on APC expression of costimulatory molecules and MHC II as well as proinflammatory (IL-17 and GM-CSF) T cell cytokine secretion. T cell proliferation is also a measure of APC function. Did the IFN- β -treated B cells from EAE stimulate more vigorous proliferation of the MOG 35-55-specific T cells?

Minor points:

(1) RE: Figure 4e. Is this most representative? The double (IL-17 and GM-CSF) positive is high in IFN- β treatment. The text reads, "...no difference was seen for T-helper cells expressing GM-CSF alone or both, GM-CSF and IL-17 (Fig. 4e). I apologize if I misunderstood.

(2) RE: style. The authors wrote, "trend toward a positive correlation." Why not write, "no clear correlation" or "no clear positive correlation"? since it was not significant.

(3) Please check for a few grammatical points. e.g. line 117, "chemokine that traffics" - might be "and CCL20, a cytokine that promotes trafficking of Th17 cells... line 259, should that be "lead to..." or either "led to..." or "leads to..."

Reviewer #1 had three main concerns and few minor concerns which we have addressed as follows:

1. *“A visual model incorporated into the figures would be highly useful. In particular, the role of different B cell subsets, the presence of B cells in the periphery vs. the CNS, and the signaling between T cells and B cells would be more clearly conveyed in a visual manner.”*

We have now added a visual representation of our hypothesis that IFN-I drives TH17 pathology by inducing IL-6 in activated B-cells (Supplemental Figure 5).

2. *“Additional information from a few experiments would be useful for more definitive conclusions. In particular, in Figure 5c a control without any interferon or anti-IL-6R in parallel would allow for the conclusion that the combination is suppressive of disease, rather than to a severity exhibited by non-interferon treated mice. Also, in the model incorporating B cells receptiveness to IFN-I in order to facilitate T cells response, expression of IFNAR by B cells would be important. Further, addition of anti-IFNAR antibodies in the experiments shown in Figure 6C & Figure 6D would greatly aid the conclusion.”*

For Figure 5c, a control group of mice treated with isotype control and vehicle has been added for comparison with interferon and anti-IL6R treatment, which highlights the suppression of disease with anti-IL6R treatment.

To address the responsiveness to IFN-I, we have also assessed the expression of IFNAR in B cells from healthy and EAE mice (Supplemental Figure 4C).

3. *“It would be appropriate to tone down assertions of prognosis in NMOSD based on this study as it does not include longitudinal data.”*

We have taken this point into consideration and have toned down the discussion of the prognostic utility of our discoveries and discussed the importance of future longitudinal studies.

4. *“Statistical considerations are important; the number of subjects included is impressive given the prevalence of NMOSD. However, how definitive are the divisions of IFN-I categories if the number of subjects is so low? An expanded discussion regarding how the findings will hold up amongst a larger cohort would help.”*

We have elaborated on the use of larger cohorts and how that would impact our findings in the discussion.

5. *“The assertion that anti-MOG and anti-AQP4 NMOSD disease states are founded on a common pathogenesis based on this study is an overstatement. This should be emphasized less.”*

We have excluded this assertion in the new version of the manuscript. In fact, we did find there to be significant differences in relapse rates in B cell depleted MOG+ and AQP4+ patients. Furthermore, levels of serum IL-6 and MCP-3 were different in MOG+ vs AQP4+ patients. These new analyses are in Figure 2 G-J and we have devoted a lengthy paragraph in the discussion to address these data.

6. *“The antigen specificity of B cells should be discussed, particularly as the underpinnings of NMOSD appears to be antigen driven. How antigen-specific B cells may exhibit particular responsiveness to IFN-I may be critical to the model.”*

We have now included a passage in the discussion on whether there could be different effects of IFN-I on self and foreign antigen-specific B cells in NMOSD and healthy individuals.

Reviewer #2 also had three main concerns and a few minor comments which we addressed as follows:

1. *“Rituximab/ocrelizumab treated patients were compared to other NMO patients, which included patients that were treated with other therapies or were not on treatment combined. Did the untreated patients as a group differ (e.g. transcriptome) from patients on treatments other than anti-CD20 treatment?”*

We compared the transcriptomes of the untreated and other-treated NMOSD patients and found no statistical differences between these groups. Furthermore, we have included untreated, other-treated and rituximab-treated patients in our transcriptomics analysis (Figure 1-d) and we still show an elevation of IFN-I genes to be a shared phenotype of all three NMOSD groups. We also

compared levels of three key serum proteins, IL-17, IL-6 and MCP-3, in untreated, other-treated and rituximab-treated patients (Fig 2F).

2. *“Both AQP4+ and MOG+ patients were included in this study. Nearly 30% were MOG+. Were there any distinguishing features between these groups? This is potentially important as there are recent publications suggesting that patients with MOG antibody-associated disease do not respond to CD20 B cell depletion as well as patients with AQP4+ NMOSD (e.g. Durozard, P., et al. Ann Neurol 2019). Also, anti-IL-6R treatment was effective in AQP4+ patients, but not AQP4- patients (Yamamura, T., et al. NEJM 2019).”*

Interestingly, we found no significant differences in the transcriptomes of the MOG+ and AQP4+ patients. However, in our cohort, we found that MOG+ NMOSD had increased relapse rates on B cell depletion therapy compared to AQP4+ patients (Fig 2H). Furthermore, the MOG+ NMOSD patient population also had relatively low levels of IL-6 and MCP-3 in their sera compared to AQP4+ patients (Fig 2I-J). We have also discussed these data in the context of the papers by Durozard and Yamamura in the discussion.

“With regard to the antigen presenting capability of B cells, for the in vitro studies shown in Fig 6, the authors focused on APC expression of costimulatory molecules and MHC II as well as proinflammatory (IL-17 and GM-CSF) T cell cytokine secretion. T cell proliferation is also a measure of APC function. Did the IFN- β -treated B cells from EAE stimulate more vigorous proliferation of the MOG 35-55-specific T cells?”

We determined how IFN-I-treated B cells from EAE and healthy mice impact the proliferation of antigen-specific T cells (Figure 6d-e & Supplemental Figure 4d). Our new data show that IFN-I treated B cells derived from EAE mice have an enhanced capacity of driving T cell proliferation (measured by Ki67 staining) compared to IFN-I treated B cells from healthy mice.

3. *“RE: Figure 4e. Is this most representative? The double (IL-17 and GM-CSF) positive is high in IFN- β treatment. The text reads, “...no difference was seen for T-helper cells expressing GM-CSF alone or both, GM-CSF and IL-17 (Fig. 4e). I apologize if I misunderstood.”*

Thank you for pointing out this typographical error. T-helper cells co-expressing IL-17 and GM-CSF were elevated in the CNS of TH17-EAE mice treated with IFN- β . We have corrected the text and the figure.

4. *“RE: style. Th authors wrote, “trend toward a positive correlation.” Why not write, “no clear correlation” or “no clear positive correlation”? since it was not significant.”*

We have changed descriptions from “trend toward” to “no correlation”.

5. *“Please check for a few grammatical points. e.g. line 117, “chemokine that traffics” - might be “and CCL20, a cytokine that promotes trafficking of Th17 cells... line 259, should that be “lead to...” or either “led to...” or “leads to...””*

We have checked for grammatical errors and corrected them.

REVIEWERS' COMMENTS:

Reviewer #2 (Remarks to the Author):

The authors have done an excellent job addressing comments of the reviewers. Congratulations on an important study.

Scott S. Zamvil

There were no concerns from the reviewer.